# Virtual staining overlay enabled combined morphological and spatial transcriptomic analysis of individual malignant B cells and local tumor microenvironments

**Zihang Fang**[1]                                                    ZFANG@PICTORLABS.AI

[1] *Pictor Labs Inc, Los Angeles, CA*

**Raymond Kozikowski**[1]                          RAYMOND.KOZIKOWSKI@PICTORLABS.AI

**Kevin de Haan**[1]                                             KEVIN@PICTORLABS.AI

**Serge Alexanian**[1]                                     S.ALEXANIAN@PICTORLABS.AI

**Michael E. Kallen**[2]                             MKALLEN@SOM.UMARYLAND.EDU

[2] *Department of Pathology, University of Maryland School of Medicine, Baltimore, MD*

**Alyssa Rosenbloom**[3]                            AROSENBLOOM@NANOSTRING.COM

[3] *Nanostring® Technologies, Seattle, WA*

**Charlie Glaser**[3]                                      CGLASER@NANOSTRING.COM

**Mark Conner**[3]                                        MCONNER@NANOSTRING.COM

**Yan Liang**[3]                                             YLIANG@NANOSTRING.COM

**Kyla Teplitz**[3]                                        KTEPLITZ@NANOSTRING.COM

**Joachim Schmid**[3]                                    JSCHMID@NANOSTRING.COM

**Jaemyeong Jung**[3]                                     JJUNG@NANOSTRING.COM

**Yair Rivenson**[1]                                      RIVENSON@PICTORLABS.AI

## Abstract

B-cell lymphomas are complex entities consisting of a component of malignant B-cells admixed in a local tumor microenvironment (TME) inflammatory milieu. Discrete characterization of both compartments can drive deeper understanding of pathophysiology, allowing more accurate diagnoses and prognostic predictions. However, limitations in both pathologist time and input tissue to generate multiple stains can greatly limit accurate identification of minute, cellular-level regions of interest necessary to achieve the full potential of spatial biology. Here, we present a novel method to perform precise sampling of cells for transcriptomic analysis using virtual staining of autofluorescence images via deep learning algorithms. We validated the performance of the model on regions of interest (ROIs) identified on chemically stained images by board certified pathologists against virtually stained images. The results confirmed the usability and accuracy of the workflow and identified distinct transcriptomic profiles across a range of virtually identified ROIs, raising the possibility of our workflow's applicability across a broader range of pathologies and tissue types.

**Keywords:** Deep learning, Histology, Digital pathology.

## 1. Introduction

B-cell lymphomas are heterogeneous diseases with respect to gene expression and tumor microenvironment, with some entities containing a minority of malignant B-cells in a largely inflammatory background. These rare malignant cells have unique interactions with the local tumor microenvironments (TME) which may drive clinical behavior and treatment responsiveness. Accurate elucidation of such interactions requires sampling of both malignant

cells and background TME for ancillary studies such as transcriptomic analysis, though existing/standard technologies cannot achieve high precision, complete spatial alignment with preservation of architectural and structural context.

Previously (Rivenson et al., 2019), it has been demonstrated that a fully supervised deep learning technique can be used to virtually stain autofluorescence images of unlabelled tissue into various stain combinations using deep learning. Here we applied the same technique and generated virtually-stained H&E images based on the autofluorescence images of the sample tissue slides. Using this novel computational technique, board certified pathologists reviewed perfectly spatially aligned virtual stains to precisely identify and annotate specific populations at the cellular-level; these areas were then seamlessly incorporated into Nanostring's GeoMx® Digital Spatial Profiler (Zollinger et al., 2020) platform allowing analysis at hitherto unrealizable levels of resolution. Analysis of the ROIs revealed distinct transcriptional profiles between areas enriched in Reed-Sternberg cells and the associated inflammatory milieu demonstrating the viability and utility of virtual H&E staining technology as a part of spatial genetic workflow.

## 2. Methods

58 unstained sections cut at 4 microns were prepared from a total of 12 Formalin-Fixed Paraffin-Embedded (FFPE) tissue blocks. These unstained slides were first scanned into autofluorescence images using four fluorescent filter cubes (BGYR) on the GeoMx® DSP platform. 17 were then deparaffinized and chemically stained with H&E whereas the other 41 underwent an additional mock GeoMx® CTA assay before chemical H&E staining. The brightfield whole slide images (WSIs) of all 58 chemically stained H&E slides were captured using a slide scanner microscope (AxioScan Z1, Zeiss) at 20x. For each tissue slide, a multi-stage registration was performed to match the brightfield (BF) WSI to its autofluorescence (AF) counterpart at subpixel level, enabling a supervised training approach. In order to learn an accurate transformation from 4-channel autofluorescence images to their corresponding brightfield H&E stained images, a conditional GAN-based (Goodfellow et al., 2014) model with an L1 loss was utilized, with a U-net architecture (Ronneberger et al., 2015) for the generator. The training framework is shown in Figure 1.

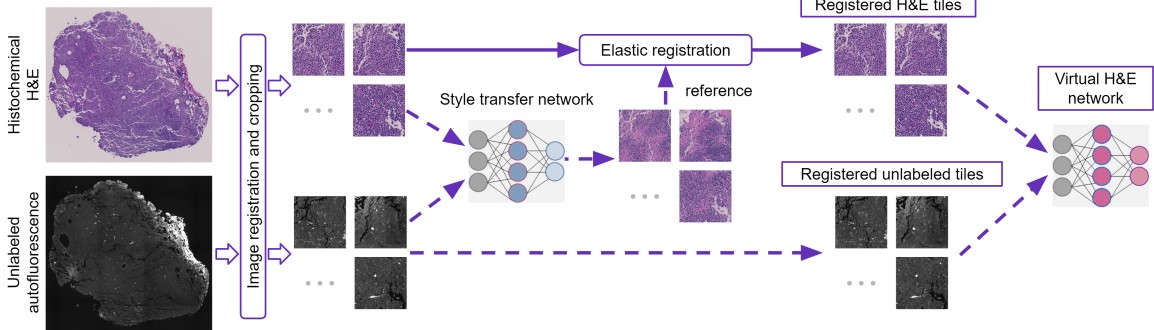

Figure 1: Training framework of the virtual stain neural network.

## 3. Results

Once the virtual H&E model was trained and validated, it was used to virtually stain autofluorescence images of unlabeled tissue from Classic Hodgkin Lymphoma (CHL) and normal/reactive lymphoid tissue not previously used for training. This virtual H&E was imported into the GeoMx® DSP platform and aligned such that any ROIs created by pathologists based on the virtual H&E would directly map to matched tissue coordinates. These selected regions were then targeted for spatially precise transcriptomic analysis. In this study, pathologists were able to identify and create ROIs for areas enriched in Reed Sternberg cells, the malignant B-cells in CHL, and separate TME regions enriched for the inflammatory milieu on the virtual H&E. Additional analysis showed clear differences in transcriptional profiles between areas as a function of the ratio of Reed Sternberg cells vs the inflammatory milieu in CHL (see Figure 2). The unstained tissue slides which were previously virtually stained were later histochemically stained by H&E for side-by-side comparison by board certified pathologists. This visual assessment further validated our virtual staining model as all selected ROIs from the virtual H&E were correctly identified and confirmed on the histochemical H&E slides.

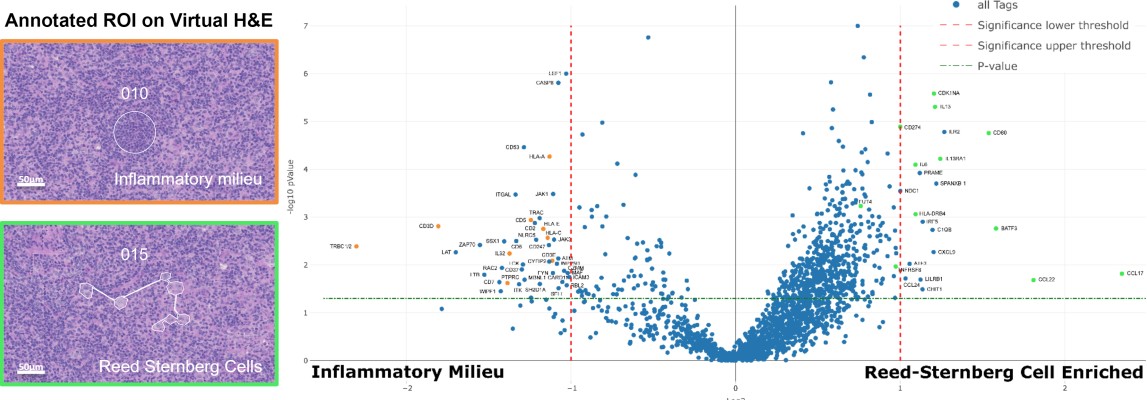

Figure 2: Volcano plot resulting from the differential expression analysis.

## 4. Conclusion

The ability to produce and review virtually stained H&E slides embedded within a downstream spatial transcriptomics pipeline greatly improves the speed, accuracy, and functionality of this complex workflow. Compared with existing processes, in which downstream analytics such as transcriptomics are deployed without careful architectural and structural context, inserting a virtual H&E annotation step enhances the ability of users to precisely define areas of interest and avoid off-target analytics. This seamless hybrid workflow increases the relevance of output transcriptomics analysis, provides accurate, real-time segmentation of different constituents of a mixed malignant/TME lesion, and preserves additional tissue. Our analysis demonstrates that the virtually stained images are concordant with chemically stained H&E slides and can be used for QC, ROI identification, and other downstream analyses. As more virtual staining models are developed, these novel capabilities will further improve the ability to precisely segment the original scanned image, unlocking increasing larger amounts of data from diminishing input tissues.

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
