# OpenReview forum: "Virtual staining overlay enabled combined morphological and spatial transcriptomic analysis of individual malignant B cells and local tumor microenvironments"
_MIDL.io/2023/Short_Paper_Track — MIDL 2023 Short paper track Poster_

### Official Review · Reviewer_5ox6 · 2023-04-10
**Combined morphological and spatial transcriptomic analysis**

**Rating:** 8
**Confidence:** 5

**Review:**

This paper performs sampling of cells for transcriptomic analysis using virtual staining of autofluorescence images via deep learning algorithms.
The advantages of the paper include:
+ This is a timely needed technology to combine morphological and spatial transcriptomic analysis
+ The results are convincing
The limitation of the paper includes:
- The patch based registration method might not lead to a globally optimal solution
- The computational efficiency is not clear

---

### Official Review · Reviewer_w1fb · 2023-04-24
**Virtual staining overlay enabled combined morphological and spatial transcriptomic analysis of individual malignant B cells and local tumor microenvironments**

**Rating:** 7
**Confidence:** 4

**Review:**

This paper presents an approach to generate virtual H&E staining from autofluorescence images that are used for spatial transcriptomics analysis.
The method relies on conditional-GAN to generate virtual H&E trained with original H&E and corresponding autofluorescence unstained data.
Authors show that using the virtual HE slide pathologists could manually annotate regions to use in downstream analysis of spatial transcriptomics.

PROS
The proposed approach seems to be effective and addresses a practical problem by optimising steps in the analysis’ pipeline.
Although based on existing work, this paper focuses on an application, B-cell lymphoma, that was not explored before, to the best of my knowledge.

CONS
The approach is not completely new as it reuse the work of Rivenson et al. 2019, and it is unclear from the paper what additional technical contribution has been brought in by the authors, besides the specific application in which it is used.
The validation is fairly limited, although the paper is very short, for the MIDL community (including myself) it might be difficult to get the value of what reported in Figure 2.
I understand that thanks to the presence of virtual HE, pathologists could manually annotate regions of interest to characterise specific cells, and therefore a fine-grained analysis with DSP can be done there.
But what is the reference standard, in other words, how to know if the achieved results are good, especially compared to other approaches (letting pathologists draw on real HE slides)?